# Observation of spatiotemporal optical vortices enabled by symmetry-breaking slanted nanograting

Pengcheng Huo[1,2,6], Wei Chen[1,2,6], Zixuan Zhang[3,6], Yanzeng Zhang[1,2], Mingze Liu[1,2], Peicheng Lin[1,2], Hui Zhang[1,2], Zhaoxian Chen[1,2], Henri Lezec[4], Wenqi Zhu[4,5], Amit Agrawal[4], Chao Peng[3]✉, Yanqing Lu[1,2]✉ & Ting Xu[1,2]✉

Providing additional degrees of freedom to manipulate light, spatiotemporal optical vortex (STOV) beams carrying transverse orbital angular momentum are of fundamental importance for spatiotemporal control of light-matter interactions. Unfortunately, existing methods to generate STOV are plagued by various limitations such as inefficiency, bulkiness, and complexity. Here, we theoretically propose and experimentally demonstrate a microscale singlet platform composed of a slanted nanograting to generate STOV. Leveraging the intrinsic topological singularity induced by $C_2$ symmetry and $z$-mirror symmetry breaking of the slanted nanograting, STOV is generated through the Fourier transform of the spiral phase in the momentum-frequency space to the spatiotemporal domain. In experiments, we observe the space-time evolution of STOV carried by femtosecond pulses using a time-resolved interferometry technique and achieve a generation efficiency exceeding 40%. Our work sheds light on a compact and versatile platform for light pulse shaping, and paves the way towards a fully integrated system for spatiotemporal light manipulation.

Vortices are ubiquitous in the natural world across all length and time scales. Examples range from quantum vortices in superfluids to singular optical beams, characterized by their non-trivial topological textures. In optics, vortices with quantized longitudinal orbital angular momentum (OAM) have been observed, studied, and utilized in a wide range of classical and quantum applications[1–16], represented by their spiral phase front around the azimuth with an undefined phase and a nullified intensity at the center. Distinct from longitudinal OAM, a new class of optical vortices in the spatiotemporal domain was recently realized by sculpturing the phase front of ultrafast light pulses[17–29], thus shedding light on a new degree of freedom for light manipulation.

Such spatiotemporal optical vortices (STOVs) are featured by transverse OAM that is perpendicular to the pulses' propagation direction, allowing the mediation of energy flow in both spatial and temporal dimensions. Despite notable progress in STOV generation, existing methods either rely on inefficient nonlinear effects[19] or bulky and complicated setups for time-frequency pulse shaping[20–27]. Recently, the possibility of using metasurfaces[30–37] that are composed of an array of subwavelength nanostructures for STOV generation was theoretically discussed[38,39], which could provide a promising and straightforward approach for spatiotemporal control of light pulses. However, considering the nanostructures' complexity in fabrication,

[1]National Laboratory of Solid-State Microstructures, College of Engineering and Applied Sciences and Collaborative Innovation Center of Advanced Microstructures, Nanjing University, Nanjing 210093, China. [2]Key Laboratory of Intelligent Optical Sensing and Manipulation, Ministry of Education, Nanjing University, Nanjing 210093, China. [3]State Key Laboratory of Advanced Optical Communication Systems and Networks, School of Electronics, Frontiers Science Center for Nano-optoelectronics, Peking University, 100871 Beijing, China. [4]National Institute of Standards and Technology, Gaithersburg, MD 20899, USA. [5]Maryland NanoCenter, University of Maryland, College Park, MD 20742, USA. [6]These authors contributed equally: Pengcheng Huo, Wei Chen, Zixuan Zhang. ✉e-mail: pengchao@pku.edu.cn; yqlu@nju.edu.cn; xuting@nju.edu.cn

experimentally generating STOVs at optical frequency in a compact and readily manufacturable platform remains an outstanding challenge.

Here, we theoretically propose and experimentally demonstrate a method of STOV generation in a microscale platform composed of a slanted nanograting. We find an isolated, zero-valued singularity point upon transmission in the momentum-frequency space when the $C_2$ symmetry (twofold in-plane rotational symmetry) and the z-mirror symmetry of the nanograting is broken, which allows generation of STOVs without requiring time-frequency-time transform or spatial position alignment of the incident light pulse. We experimentally observe that the STOVs carrying transverse OAM can be generated with efficiencies exceeding 40% by using femtosecond laser pulses with a duration time of approximate 80 fs centered at a wavelength of 800 nm. Such a method significantly simplifies the setup and promotes the efficiency of STOV generation, thus paving the way for sophisticated control of ultrafast pulses using an integrated platform.

## Theory and principle

We start by considering an ultrafast light pulse propagating through an optical device which is schematically shown in Fig.1a. The device is comprised of a one-dimensional (1D) slanted grating fabricated of Si ($n_{Si}$ = 3.72) on a low-refractive-index SiO$_2$ substrate ($n_{SiO_2}$ = 1.46) that is periodic along the x-direction and has sufficient length along the

y-direction, with a slant angle denoted as $\theta$. The nanograting supports transverse polarized (i.e., along $E_y$) optical resonances in the vicinity of the second-order $\Gamma$ point ($k_x = 0$) (Fig. 1b), in which they support anti-symmetric TE-A modes (inset of Fig. 1b). For $\theta = 0°$, such a system restores $C_2$ symmetry, the TE-A mode at the Brillouin zone (BZ) center shows a bound state in the continuum (BIC) with infinite quality ($Q$) factor (Fig. 1c). With an increase of $\theta$, the system breaks the $C_2$ symmetry, transforming the BIC to a mode with finite values of $Q$ factor, thus unlocking channels for exchanging energy with the external wave.

When passing through the nanograting, the wavefront of the incident pulse is partly sculptured by the dispersion relation of the optical resonance. Specifically, the interaction between the pulse and the nanograting can be decomposed as the combination of two processes: part of the pulse energy directly transmits through the nanograting without interacting with the resonant mode, while the other part excites the resonance and radiates outward. Such processes can be depicted from the temporal coupled-mode theory (TCMT)[40–44]: the incident pulse would in principle interplay with two distinctive resonances at $\pm k_x$ with 4 ports associated with leaky rates of $\gamma_{1-4}$ shown in Fig. 1d, and the Fabry-Perot background described by transmission coefficient $t$ and reflection coefficient $r$ (see Supplementary Section 1 for details). Then, the features of the overall transmitted light field $g(\omega, k_x)$ in the frequency-momentum space can be conveyed to the pulse $G(\tau, x)$ in time and spatial domain that we observe eventually in

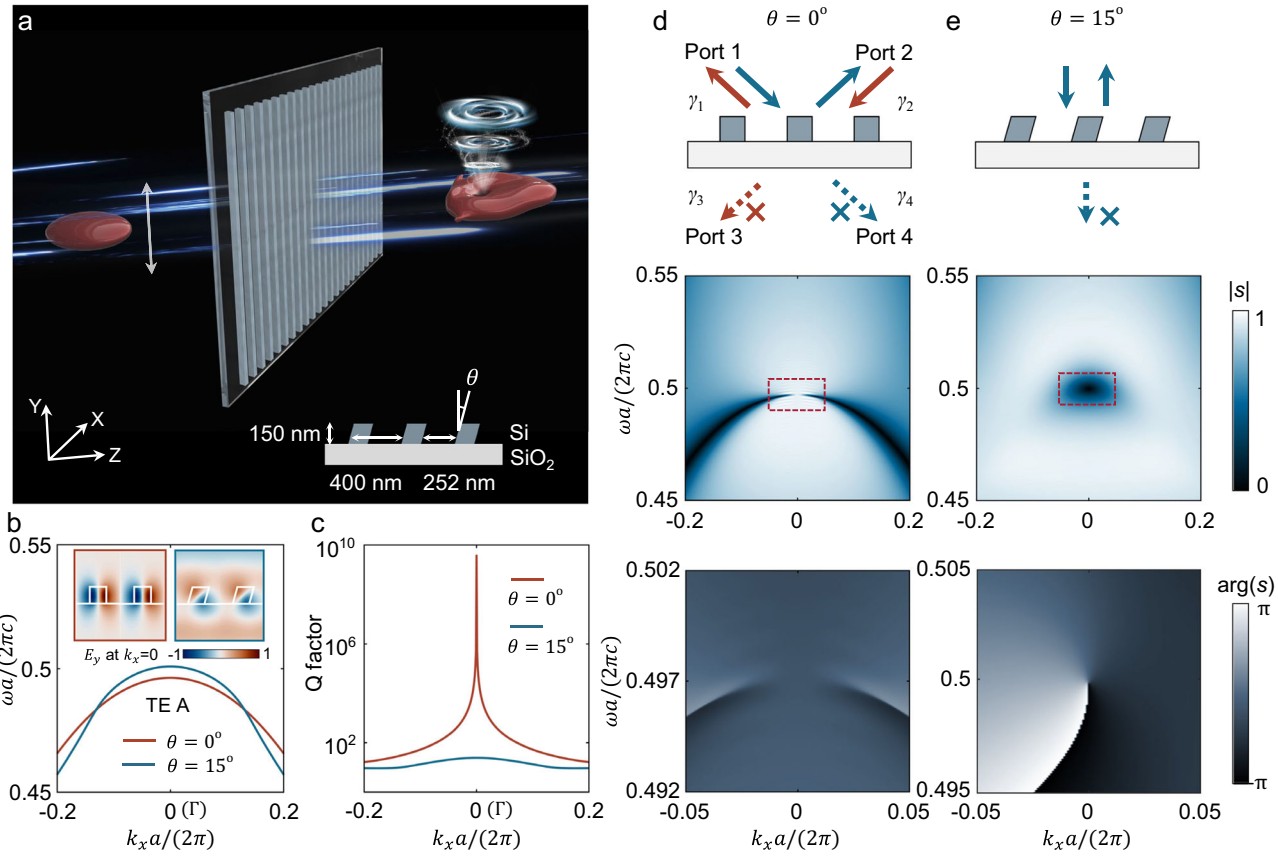

**Fig. 1 | Origin of topological singularity induced by symmetry breaking.**
**a** Schematic of STOV generation, showing an ultrafast light pulse transmitting through an optical device that is comprised of a one-dimensional slanted grating. The white arrow indicates the polarization direction of the incident pulse. **b, c** The band structure and $Q$ factors of the TE-mode along the $k_x$ axis at $\theta = 0°$ (red) and $\theta = 15°$(blue), respectively, with the insets showing the mode profiles in the two unit-cells at the $\Gamma$ point ($k_x = 0$). **d** For a $C_2$ symmetric structure, the complex

transmission coefficient $s$ presents only one nodal line in the $\omega$-$k_x$ space and does not show a spiral phase in the frequency-momentum space. **e** By breaking the $C_2$ symmetry and the z-mirror symmetry, an isolated zero-transmission singularity associated with a spiral phase appears at the BZ center near the resonant frequency. Middle panel of (**d**, **e**): the amplitude distribution of the transmission coefficient. Lower panel of (**d**, **e**): the phase distribution corresponding to the red rectangle region.

the far-field through the two staggered Fourier transform relation as $G(\tau, x) = \mathcal{F}\{g(\omega, k_x)\}$, indicating that the field with a spiral phase in the $\omega$-$k_x$ domain can be transformed into a field with a spiral phase in the $\tau$-$x$ domain which is exactly the STOV we are pursuing. Recall that the electromagnetic field is nullified at the vortex center, so the condition of generating STOV is equivalent to finding the zero-valued singularities in $g(\omega, k_x)$.

By employing the two-resonance TCMT model (see Supplementary Section 1 for details), the conditions of singularity in complex transmittance can be concluded as:

$$\begin{cases} (\omega - \omega_0)^2 = \frac{4r^2\gamma_2\gamma_4}{t^2} - (\gamma_2 - \gamma_4)^2 \\ \gamma_1 = \gamma_2 \\ t \neq 0 \end{cases} \tag{1}$$

which is naturally fulfilled at any $k_x$ with the protection of $C_2$ symmetry or $z$-mirror symmetry (see Supplementary Section 2 for details). As verified, we obtain a nodal line in transmittance spectrum in the $\omega$-$k_x$ space (Fig. 1d, middle panel). However, since such singularities are not isolated, they cannot exhibit as a spiral phase in the $\omega$-$k_x$ space (Fig.1d, lower panel) and thus cannot be utilized to generate STOVs.

In order to find an isolated singularity in transmittance to generate a STOV, we turn to a nanograting with broken $C_2$ symmetry and $z$-mirror symmetry, for instance, using the slanted grating on a low-index substrate as shown in Fig. 1e. At nonzero $k_x$ point, the singularity

conditions in Eq. (1) are ruined (typically $\gamma_1 \neq \gamma_2$) so that the nodal line would disappear. However, at the time-reversal invariant momentum $k_x = 0$, the condition $\gamma_1 = \gamma_2$ still preserves to result a topologically robust isolated singularity in the $\omega$-$k_x$ space. Confirmed by numerical simulations (Fig. 1e, mid panel), a vanishing point of transmisstance emerges at the BZ center with a slight frequency detuning from the eigenfrequency of mode TE-A. Clearly, such a singularity creates a spiral phase in the $\omega$-$k_x$ space (Fig. 1e, lower panel), which readily corresponds to a STOV in the $\tau$-$x$ domain.

## Design of the slanted nanograting

According to the discussion above, we present a detailed design of a slanted nanograting for STOV generation. To best capture the nature of realistic samples, we add into our design an optical extinction coefficient term for Si material ($k_{Si} \approx 0.0065$) based on ellipsometry measurements to depict the non-radiative loss $\gamma_{nr}$ that is contributed by absorption and scattering, and then we derive the TCMT model accordingly (see Supplementary Section 3 for details). The theory shows that the zero-valued singularity still exists but it slightly deviates from the BZ center as $k_xa/2\pi = -0.0057$, calculated by numerical simulations as shown in Fig. 2a.

We further investigate the evolution of singularity under different slant angles $\theta$ in the $\omega$-$k_x$ space. Assuming the non-radiative loss is constant $vs.$ $\theta$, the singularity gradually moves away from the $\Gamma$ point when $\theta$ decreases from 30° to 2°. Note that the singularities and the

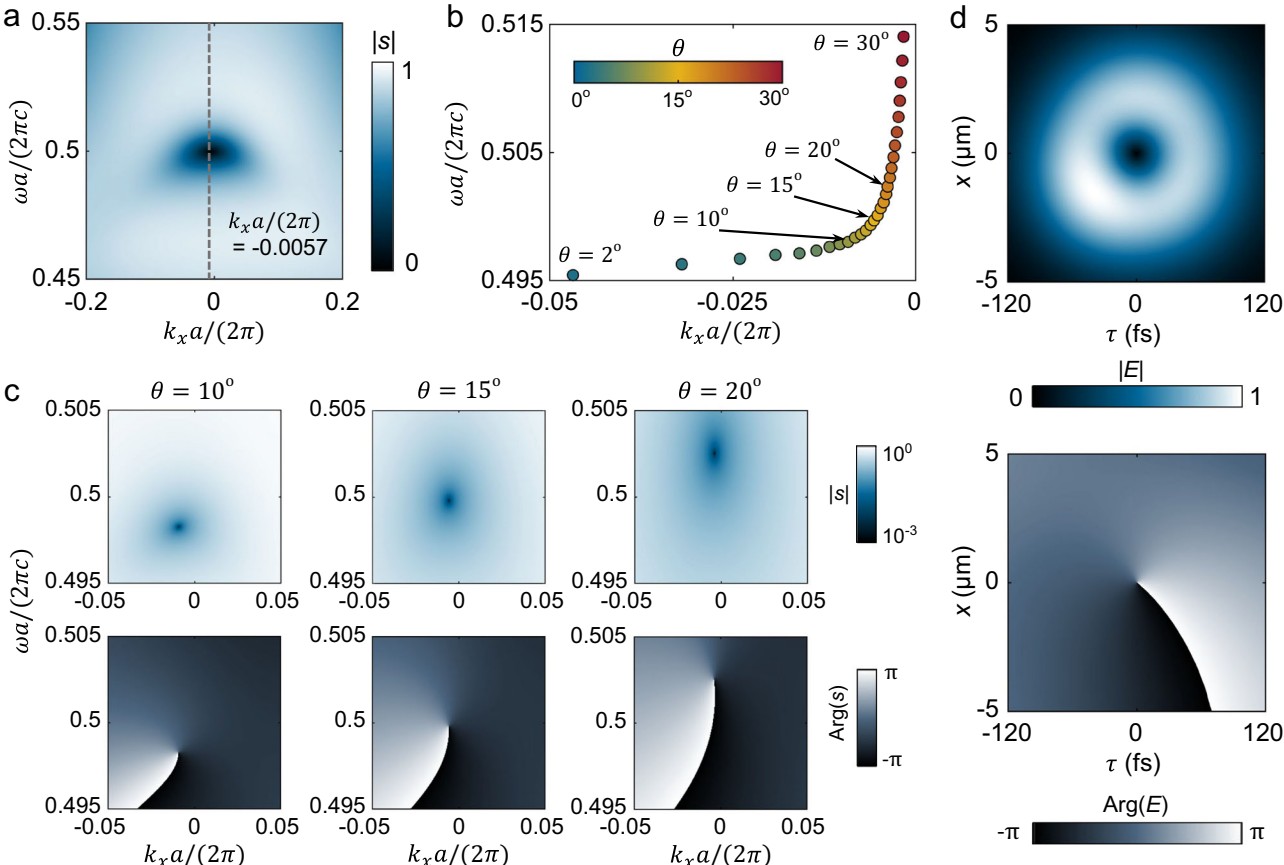

**Fig. 2 | Design of the device with taking non-radiative loss into account. a** By adding an optical extinction coefficient to the material ($k_{Si} = 0.0065$) to represent non-radiative losses, the topological singularity of the transmission coefficient does not vanish but deviate to an off-$\Gamma$ position in $\omega$-$k_x$ space. **b** The evolution of the topological singularity in $\omega$-$k_x$ space when the slant angle of grating $\theta$ varies from

30° to 2°. **c** The corresponding complex transmission coefficient distribution in the frequency-momentum space for slant angles $\theta = 10°$, 15°, and 20°. **d** The amplitude and phase distribution of the transmitted STOV in the spatiotemporal domain after a Gaussian pulse propagates through the nanograting with the slant grating angle of $\theta = 15°$.

associated spiral phase patterns always exist and continuously evolve across all the slant angles (Fig.2b), proving that the singularity is topologically robust in a 2D parameter space. However, a large slant angle reduces the transmittance of the device (Fig.2c), which would lower the efficiency of the generated STOV. Therefore, we choose a moderate value of $\theta = 15°$ for experimental verification. To directly show the feasibility of STOV generation, we simulate a compressed Gaussian pulse with a central carrier frequency of 374.5 THz, nominal beam waist of 3 μm, and pulse duration is 80 fs that passes through the designed device. The monitor is positioned 2 μm away from the device. As shown in Fig.2d, the transmitted pulse exhibits a unique donut pattern in the spatial-temporal, $\tau$-$x$, domain. Clearly, the donut-like field distribution shows zero amplitude at its center and carries a phase vortex with a topological charge of $l = +1$, indicating the emergence of STOV that carries transverse OAM.

### Experimental generation of STOV

To experimentally verify the proposed method of STOV generation, we fabricated a slanted nanograting on a thin-film of Si on a fused-silica substrate, where the slanted grating with $\theta = 15°$ is fabricated using plasma-enhanced chemical vapor deposition, electron-beam lithography, lift-off and customized reactive ion etching processes (see details in Methods and supplementary Fig. S1). The key to fabricating

the slanted grating side walls is to use a wedged substrate instead of the conventional planar substrate during the dry etching, which allows us to precisely control the slant angle of the grating. Figure 3a presents an optical microscope photograph of the fabricated sample with a nominal size of 30 μm × 50 μm. Such a size is designed primarily to demonstrate the non-alignment characteristics of STOV generation, but in principle, the device footprint could be further shrunk to a few micrometers. The device exhibits a very uniform grating structure with a measured slant angle of ≈ 15°, as shown in the tilted-view (Fig. 3b) and side-view (inset of Fig. 3b) of scanning electron microscope (SEM) images. The angle-resolved transmission spectra of the device under transverse-electric (TE) polarized illumination are measured and presented in Fig. 3c, which agree well with the theory and simulations shown in Fig. 2a. The transmission spectra possess a clear minimum centered at $k_x a/2\pi \approx -0.0105$ and $\omega a/2\pi c = 0.5$ in the 2D $\omega$-$k_x$ space (Fig. 3d), forming an isolated, zero-valued singularity that is required for the generation of STOV.

A time-resolved scanning interferometry setup is built for observing and characterizing the STOV generated from the slanted nanograting (Fig. 4). An ultrashort femtosecond pulse with a center wavelength of 800 nm and a pulse duration of ≈80 fs (see Supplementary Section 4 and Fig. S3 for details) is generated by a Ti-sapphire amplifier and then sent through a polarizer oriented along the $y$-direction. Further, the

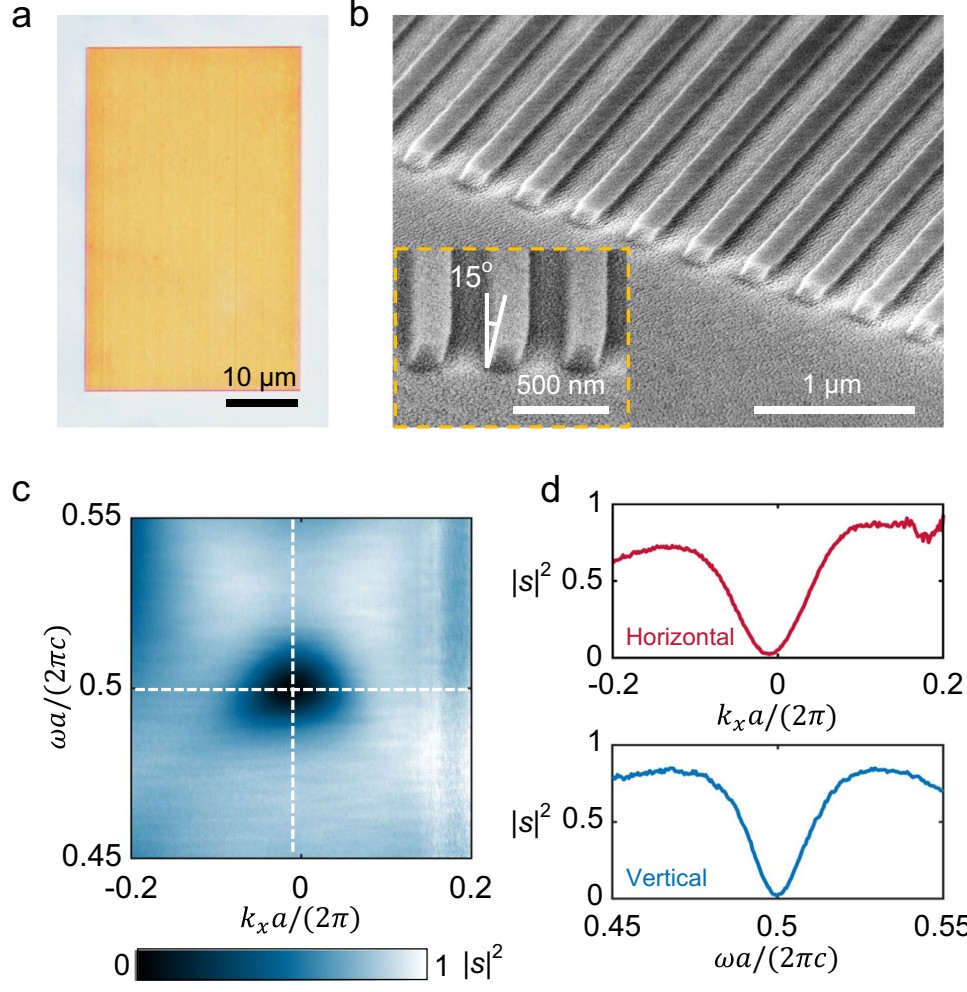

**Fig. 3 | Fabrication and characterization of slanted nanograting. a** An optical microscope photograph of the fabricated slanted nanograting. **b** A tilted-view and a side-view scanning electron microscope images of the sample. **c** The angle-dependent transmission spectra of the device measured under TE polarized incidence. **d** Cross sections of the transmission spectra along horizontal (top) and vertical (bottom) white dashed lines in (**c**).

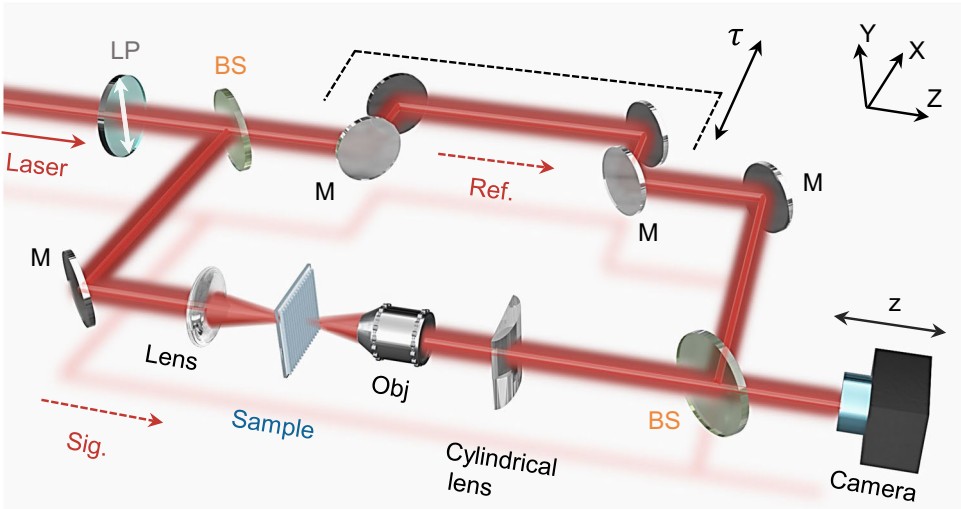

**Fig. 4 | Schematic of the experimental setup for characterizing the STOV.** A femtosecond laser pulse with a duration time of ≈80 fs centered at a wavelength of 800 nm is sent into the measurement system (labeled as "Laser"). The signal beam (labeled as "Sig.") transmits through the slanted nanograting to generate STOV. The reference beam (labeled as "Ref.") is delayed by a tunable time of τ with respect to the signal beam to perform interference. The interference fringes are recorded by a CCD camera. To make the drawing clearer, some mirrors for zero delay are omitted. LP linear polarizer, BS beam splitter, M mirror, Obj objective.

incident pulse is split into a reference pulse and a signal pulse and they travel in two parallel light paths respectively. The reference pulse is controlled by a translation stage and movable mirrors to precisely tune the delay time, τ. While the signal pulse is focused and transmits through the device sample, a $4f$ optical imaging system along the $x$-axis that consists of an object lens and a cylindrical lens, and further combines with the reference pulse by using a beam splitter to form interference fringes at the camera. By tuning the delay time τ, such a setup allows us to accurately record the spatial phase distribution at each temporal slice, enabling us to observe the fine structure of the STOV.

In the experiment, we first focus the incident pulse at an arbitrary position P1 on the sample as shown in Fig. 5a, left panel, in which the dashed box represents the boundary of the fabricated sample. Since the incident pulse spread on a range of $k$ that is determined by the numerical aperture (NA) of the focusing lens, we expect its transmittance to be modulated by the dispersion relation of the resonant mode in $\omega$-$k_x$ space. Subsequently, the spatiotemporal characteristics of the transmitted pulse are recorded by a charged coupled device (CCD) camera located at the focal plane (designated as the location, $z = 0$ mm), after passing through the cylindrical lens. As shown in Fig. 5a, right panels, the recorded data comprises five consecutive fringe patterns, with a delay time interval of ≈ 80 fs between each recording. The interference fringes are observed to be smooth and continuous at the beginning and end of the time delay (τ ≈ −160 fs and 160 fs). As the delay τ approaches 0, the left and right fringes gradually bend in opposite directions and eventually form a series of forked patterns on the central axis, indicating the presence of a spatio-temporal phase singularity. The experimentally observed phase behavior agrees very well with theoretical fringe patterns (Fig. S5).

By scanning the time delay τ continuously, the complete spatio-temporal profile of the transmitted pulse is computationally reconstructed from the delay-dependent interference fringes. The reconstructed intensity and phase distribution of the pulse wave envelope at $z = 0$ mm are presented in the middle panels of Fig. 5b. The wave envelope is shown to be donut-shaped with a spiral phase singularity of topological charge $l = 1$, clearly evidencing that the transmitted pulse carries a STOV. To further investigate the propagation dynamics of the STOV, the transmitted pulse wave envelope is also reconstructed at the locations $z = -45$ mm and $z = 45$ mm, as illustrated in the left and right panels of Fig. 5b. The evolution of the STOV is

symmetric in time and exhibits two separated lobes in the spatial-temporal diagonal, mainly due to time diffraction effects of transverse OAM beams[27]. The time diffraction effect describes the spontaneous evolution of STOVs during free-space propagation, resulting from the accumulation of different phase shifts by various temporal frequency components within the STOV.

Because the singularity is in frequency-momentum space and the structure is almost homogeneous, we expect that the incident pulse does not need to be accurately aligned to a particular spatial location on the sample, which significantly relaxes the complexity of optical alignment. Accordingly, we shift the sample in the $x$-$y$ plane that makes the incident pulse focus at two different positions P2 and P3 on the sample, as shown in Fig. 5c, left panel. Following the same measurement approach mentioned above, we observe that the transmitted pulse wave envelope still shows unique characteristics of STOV at $z = 0$ mm (the corresponding spatiotemporal evolutions at different $z$ positions are given in supplementary Fig. S6). The average efficiency of the STOV generation from P1 to P3 is measured to be approx. 42% in the experiment. The generation efficiency can potentially be improved by either broadening the spectrum of the incident pulse or decreasing the slant angle of the gratings constituting the device. In addition, we also envision some possibilities for using the platform to modulate STOVs, especially to generate higher-order STOVs. Firstly, we note that the OAM direction of the STOVs can be conveniently controlled using a few simple operations on this platform. For instance, rotating the device 180° around the $z$-axis can reverse the transverse OAM direction. Secondly, benefiting from the design and processing advantages of ultrathin planar structures, the STOVs of high-order topological charges can be realized by configuring multiple layers of slanted grating (Fig. S7).

In summary, we propose and demonstrate a new method of direct STOV generation in free-space by utilizing a slanted nanograting. By breaking the $C_2$ symmetry and $z$-mirror symmetry, an isolated, zero-valued singularity of transmittance raises in the frequency-momentum space that carries a spiral phase around it. The singularity is topologically robust in the real space, and can readily transform to STOV through Fourier optics. We design and fabricate an optical device that comprises a one-dimensional slanted grating, and observe the STOV by using an ultrafast time-resolved interferometry setup. Compared with the previous theoretical discussion[35,38,39], we further optimize

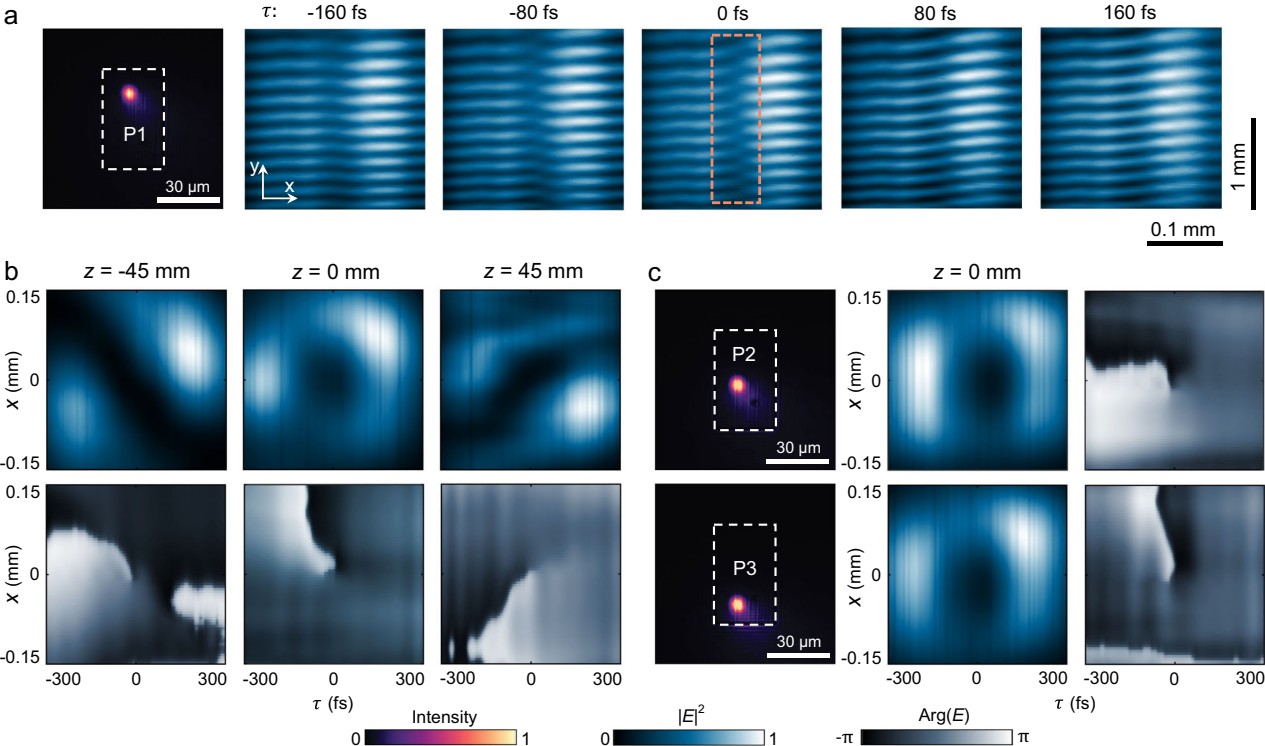

**Fig. 5 | Observation and characterization of STOV. a** Five representative fringe patterns at an interval time of ≈80 fs are recorded in the *x*-*y* plane corresponding to the STOV generated at position P1 in real space. The white dashed box denotes the boundaries of the sample. The forked pattern that indicates the presence of a phase singularity is highlighted by an orange dashed box. **b** Reconstructed intensity and phase distribution of the transmitted pulse in the spatiotemporal domain at different locations of $z = -45$ mm, 0 mm, and 45 mm, showing the evolution of STOV along the propagation direction. **c** Reconstructed intensity and phase distribution of the STOV formed at positions P2 and P3, proving the alignment-free nature of STOV generation.

device design and experimental setup, and demonstrate a generation efficiency exceeding 40% by using femtosecond laser pulses with a duration time of ≈ 80 fs centered at a wavelength of 800 nm. Our method significantly simplifies the complexity of STOV generation and paves the way towards an integrated system for ultrafast pulse shaping, thus broadening the horizon of applications leveraging spatiotemporal light manipulation.

## Methods
### Sample fabrication
The proposed slanted nanograting is fabricated by using a series of processes including plasma-enhanced chemical vapor deposition (PECVD), electron-beam lithography (EBL), lift-off, and reactive ion etching (RIE). The step-by-step fabrication process is illustrated in Fig. S1. First, a 150 nm thick layer of amorphous silicon (α-Si) is deposited on a 500 μm thick fused silica substrate using PECVD. The refractive index of α-Si is measured using spectroscopic ellipsometry and shown in Fig. S2. The α-Si surface is then cleaned with oxygen plasma to enhance adhesion, after which a 200 nm thick positive-tone electron-beam resist is spin-coated at 4000 rotations per minute on the α-Si film. The sample is then baked on a hot plate for 3 min at 180 °C. To mitigate the charging effect during the EBL process, an anti-charging conductive polymer is spin-coated at 4000 rotations per minute on the resist, and baking is done for 90 s at 90 °C. Next, the designed patterns are defined in the resist by an electron beam lithography system at an accelerating voltage of 30 keV, and development is implemented in hexyl acetate. Afterward, a 30 nm thick Al layer is deposited on the resist using an electron beam evaporator, and a lift-off process is immediately performed in n-methyl-pyrrolidone (NMP) at 80 °C. To control the slant angle of the gratings precisely, a wedged

holder is used instead of a planar substrate during the etching process. The sample is placed on a customized wedged holder made of fused silica with a slanted angle of 15°. Reactive ion etching using $CF_4$ chemistry is then performed to transfer the patterns from Al to the α-Si layer. Finally, the desired sample is obtained by removing the residual aluminum mask with an Al etchant. The target width of the air gap is achieved by fabricating a series of samples with the parameter swept from 230 nm to 270 nm at a fixed slant angle.

### Spectral measurement
The angle-dependent transmission spectra of the devices are measured with an angle-resolved spectroscopy microscopic system (IdeaOptics Instruments). The incident light polarization is set to along the y direction (TE). The transmitted light is collected by a 100× near-infrared microscope objective lens with a numerical aperture of 0.95. By applying an aperture in the optical path, the illumination area is limited to be 15 μm × 30 μm approximately. In experiments, the spectra resolution is 0.75 nm, and the angular resolution is 0.2 degree.

### Optical characterization
To extract time-resolved intensity and phase information from the generated spatiotemporal optical vortex (STOV), we utilize Mach-Zehnder scanning interferometry. Reference pulses, at a center wavelength of 800 nm and a duration of ≈ 80 fs, are obtained from the initial laser pulse through a beam splitter. The STOV and reference beam follow different optical paths and recombine at another beam splitter, where a motorized translation stage with a step of approx. 1.2 μm adjusts the reference path length. When the STOV and the reference pulse overlap, interference fringes form and are recorded by a charged coupled device (CCD) camera. Note that the overlapping

reference and signal pulses maintain a slight angular tilt in the x-direction, and we fine-tuning this angle to optimize the period of the observed interference fringes in the y-direction. Since the STOV is much longer than the reference pulse, its full intensity profile is reconstructed by collecting its temporal slices using the reference pulse as it is scanned in the time domain. The measured interference patterns also enable the reconstruction of the phase information of the STOV.

More specifically, for each time delay $\tau$, we apply a one-dimensional Fourier transform along the y-axis to the interference patterns. Therefore, the one-dimensional phase profile $\phi(x)$ at each $\tau$ could be obtained by extracting the phase at the peak in the Fourier domain. We then stack the phase profiles obtained at various time delays to construct the complete phase distribution $\phi(x, \tau)$ in the spatiotemporal domain. It is important to note that instabilities in the interferometer introduce additional frame-to-frame noise between the phase profiles at different time delays. To mitigate this issue and obtain the correct phase information, we carefully apply a series of phase shifts to the phase profiles at each time delay. The objective of these phase shifts is to maintain a constant phase value at a reference point $x_0$ across all profiles, i.e., $\phi(x_0, \tau) \equiv$ constant for all $\tau$.

## Data availability
All the data in this study are provided within the paper and its supplementary information. The data of this study are available from the corresponding author upon request.

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

## Acknowledgements

We acknowledge support from the Key Research and Development Program of the Ministry of Science and Technology of China (2022YFA1205000 to T. X. and 2022YFA1207200 to P. H.), National Natural Science Foundation of China (12274217 to T. X., 62105142 to P. H., 62305157 to W. C.), the Natural Science Foundation of Jiangsu Province (BK20220068 to T.X. and BK20212004 to Y.L.). W.Z. acknowledge support under the Cooperative Research Agreement between the University of Maryland and NIST-PML, Award no. 70NANB10H193. The authors acknowledge the technique support from the microfabrication center of the National Laboratory of Solid-State Microstructures.

## Author contributions

P. H., Y. L. and T. X. conceived the idea. P. H., Z. Z. and C. P. performed the theoretical analysis and simulations. P. H., Y. Z., P. L., M. L. fabricated the metasurface samples. P. H. and W. C. performed the experimental measurements. P. H., H. Z., Z. C., W. Z., H. L. and A. A. analyzed the data. All the authors discussed the results and contributed to the manuscript.

## Competing interests

The authors declare no competing interests.

## Additional information

**Peer review information** : *Nature Communications* thanks Marcus Ossiander, and the other, anonymous, reviewer(s) for their contribution to the peer review of this work. A peer review file is available.

