## [Peer Review File · Nature Communications]

Observation of spatiotemporal optical vortices enabled by symmetry-breaking slanted nanogratingREVIEWER COMMENTS

Reviewer #1 (Remarks to the Author):

Review for “Observation of spatiotemporal optical vortices enabled by symmetry-breaking nonlocal metasurfaces” by Huo et al.

In their paper, Huo et al. report on novel slanted metasurfaces/gratings that can create optical spatiotemporal vortices. The symmetry-breaking via slanting is intriguing and the claimed application would be a great and timely proof demonstrating the concept and its power.

Unfortunately, I believe that currently the data does not support the claim that a spatiotemporal vortex has been produced:

The manuscript does not present a characterization of the spectral/temporal phase of the ultrafast pulse in the reference arm and judging from Fig. 4, the arms have extremely imbalanced dispersion. As the presented spatial interferometry measurement is a linear measurement, it measures the phase difference between the reference pulse and the signal pulse.

Therefore, without a known reference, I believe the presented measurements can prove that the metasurface creates a certain momentum-time phase profile (which, however, is currently tainted by the dispersion imbalance between the interferometer arms). However, they cannot prove that pulses with the claimed momentum-time profile were produced. If I am not mistaken, a measurement resembling the data presented in Fig. 5 could have been taken with a spatially coherent, but temporally incoherent light source, too. Furthermore, the manuscript methods are extremely short regarding the treatment and processing of experimental data.

For the above reason, I cannot recommend the publication of the manuscript in Nature Communications in its current form. However, I would re-review the manuscript after a significant rework.

General remarks:

Most of the manuscript mentions C2 symmetry. However, in line 64 of the manuscript, also the z-mirror symmetry is mentioned. A definitive statement of the symmetries that must be broken for the described effect to occur and a figure panel illustrating these symmetries would help my understanding.

Showing the reference pulse spectrum and spectral phase and correcting for this phase is required. Showing the reference pulse spatial profile would also help evaluate the data.

The abstract, introduction, and conclusion praise this paper’s developments as a way toward a fully on-chip integrated system for spatiotemporal light manipulation. However, no part of the manuscript deals with integrated photonics. Furthermore, the required light incidence and grating directions are not compatible with integrated photonics manufacturing techniques. I think this statement is wrong or needs further discussion.

An in-depth proofing of the article usage in the manuscript, adding explanations for specialized terms, and longer explanations for mathematical deductions would help my understanding.

Specific remarks:

Line 24: 'frequency-momentum space to the spatiotemporal domain': I believe the order 'momentum-frequency space to match spatiotemporal domain' would be more correct.

Line 31: superfluid is maybe missing a noun.

Line 47: is a slanted metasurface necessary, or could one use other ways to break symmetry (e.g., how more common metasurfaces realize BICs) and still obtain the same effect?

Line 48 (reoccurs later): I believe 'fabricated on Silicon' should read 'fabricated of/from silicon'.

Fig. 1a: the coordinate system does not match between panel a and the manuscript.

Fig. 1: the polarization could be indicated with an arrow.

Fig. 1b: Gamma should be marked if it is used in the manuscript.

Fig. 1b: it would be interesting to see at least two unit cells to judge periodicity.

Fig. 1d: I is not introduced in the caption.

Fig. 1d/e: the red rectangle is not discussed in the caption.

Fig. 1d/e: The lower panels would be easier to interpret as two-dimensional false-color plots with axes labels and ticks. The third axis and tilted view add no extra information.

Fig. 1d/e, 2 c/d, 5 b/c: the employed non-cyclic colormap suggests a phase discontinuity. A cyclic colormap would mitigate this impression.

Line 61: the Gamma point should either be marked in the figure, or one could use $k_x=0$ instead.

What is the difference between a second-order Gamma point and a first-order Gamma point?

Line 64: see general remarks. Is breaking both symmetries necessary or would breaking one symmetry be sufficient?

Line 70: two staggered Fourier transforms would read more clear

Line 71: I cannot understand the sentence 'Such a relationship...'. Maybe it requires more explanation.

Line 74: can a singularity be not zero-valued? If not, I would drop the zero-valued (occurs again later in the manuscript).

Line 78: why do you expect the cancelation of the two contributions? The two could have different strengths. Or are equal strengths guaranteed somehow?

Line 82: do you mean 'for a system without asymmetry'?

Line 83: r, t should be defined, even if they are used commonly.

Fig. 2c: the lower panels would be easier to interpret as two-dimensional false-color plots with axes labels and ticks. The third axis and tilted view add no extra information.

Fig. 2c: caption says 30° , figure says 20° .

Fig. 2d: judging from the discussion of the experiment, the tau-x distribution occurs after a 4-f imaging system and in a precise position along the propagation direction. Here, it seems the distribution occurs immediately after the metasurface. If further propagation was included, this should be mentioned.

Line 118: It should either be mentioned that the pulse is compressed, or the pulse's temporal/spectral input phase should be given.

Line 118: I expect that the spatial profile of a 3 μm waist at 800 nm wavelength experiences quite the evolution upon propagation. Therefore, the position along the propagation direction at which

the beam has a 3 μm waist and the position at which Fig. 2c is recorded should be mentioned.

Fig. 3c: line-outs along the dashed lines would enable judging data quality.

Fig. 3c: no details on this measurement are provided.

Fig. 4: the polarization could be added as an arrow.

Fig. 4: the drawn interferometer does not allow zero delay. If mirrors were omitted for the purpose of a cleaner drawing this should be mentioned in the caption.

Fig. 4: the drawn interferometer arms have extremely unbalanced dispersion.

Fig. 4: the system is only 4f-imaging along the x-axis. This should be mentioned in the manuscript.

Line 140: charactering should probably read characterizing.

Line 142: same as before: it should either be mentioned that the pulse is compressed, or the pulse's temporal/spectral input phase should be given. Just giving a pulse duration and no information about the phase does not characterize the laser pulse. This should be accompanied by the measured spectral or temporal phase, using, e.g., FROG, SPIDER, or a related technique.

Line 146: the system is only 4f-imaging along the x-axis.

Line 168: Is a 90° turn between the spatiotemporal profiles $z=-45\text{mm}$ and $z=45\text{mm}$ indicative of a symmetric or an antisymmetric time evolution?

Line 170: I am unfamiliar with the term 'pure time diffraction', maybe an explanation would help.

Lines 171-177: For my interpretation of the term non-local, this line of reasoning does not prove non-locality. It proves that the metasurface/grating is the same in multiple spots. If I look at multiple spots in the same grating, I expect the same performance. A strong non-locality could maybe be shown for different-sized metasurfaces or different input beam waists?

Fig. 5a: I see forks. Data evaluation would be easier if a zoomed version of a fork and the reference and signal spatial profile without interference were also shown. Furthermore, understanding the displayed data would maybe benefit from giving a simulated (perfect) example of how the data should look.

Why are there fringes at $\tau=0$ in the y-direction? Is one wave impinging at an angle?

Methods:

Line 345-350: There is too little detail to understand or evaluate the phase extraction procedure.

Supplementary Information:

Line 35: are r , t just numbers? If they are reflection/transmission coefficients they should be called like that.

Line 69: how does s relate to S_{14} ?

Line 70/84: writing the complex equation like this adds no information.

Line 82: a bracket is missing

Line 86/88: are $-2\pi/-1$ also an option?

Reviewer #2 (Remarks to the Author):

The authors report the observation of spatiotemporal optical vortices (STOV) enabled by symmetry-breaking nonlocal meta surfaces. Recently there have been strong interests in spatiotemporal optical vortices due to the transverse orbital angular momentum carried by these novel optical fields. The authors demonstrate the feasibility of generating STOV using a compact device called meta surface. The authors use a standard interferometric method to measured the generated STOV. Given the rapidly increasing researches in STOVs, such a devices would be of interests to researchers in the field. However, there are a few points need to be addressed to justify its significance to be published in a high profile journal like Nature Communications.

1. The concept of generating STOVs using integrated devices is not new, for examples, references 35 & 38. Although these references did not report experimental realization using the proposed structures, it seems the essential concept has been revealed. Thus it is necessary for the authors to clearly identify the advantages/differences of the proposed meta surface.
2. Of course, the main technical achievement is the experimental generation of observation of the STOV generated with fabricated device. It appears that the device is just a grating with slant angle. The questions is, why the authors call this devices as meta surface? In general, meta surface rely on the geometric phase imparted by circularly polarized illumination. It doesn't appear to be the case in this paper.
3. Although as claimed by the authors that such a device is much more compact compared with the existing pulse shaper based method, the use of such device also have significant drawbacks. The main problem is the lack of flexibility. For example, can the authors generate high order STOVs, or perhaps Bessel STOV with the proposed platform?

Response to the referees' comments:

Reviewer #1 (Remarks to the Author):

Review for “Observation of spatiotemporal optical vortices enabled by symmetry-breaking nonlocal metasurfaces” by Huo et al. In their paper, Huo et al. report on novel slanted metasurfaces/gratings that can create optical spatiotemporal vortices. The symmetry-breaking via slanting is intriguing and the claimed application would be a great and timely proof demonstrating the concept and its power. Unfortunately, I believe that currently the data does not support the claim that a spatiotemporal vortex has been produced:

The manuscript does not present a characterization of the spectral/temporal phase of the ultrafast pulse in the reference arm and judging from Fig. 4, the arms have extremely imbalanced dispersion. As the presented spatial interferometry measurement is a linear measurement, it measures the phase difference between the reference pulse and the signal pulse. Therefore, without a known reference, I believe the presented measurements can prove that the metasurface creates a certain momentum-time phase profile (which, however, is currently tainted by the dispersion imbalance between the interferometer arms). However, they cannot prove that pulses with the claimed momentum-time profile were produced. If I am not mistaken, a measurement resembling the data presented in Fig. 5 could have been taken with a spatially coherent, but temporally incoherent light source, too. Furthermore, the manuscript methods are extremely short regarding the treatment and processing of experimental data.

For the above reason, I cannot recommend the publication of the manuscript in Nature Communications in its current form. However, I would re-review the manuscript after a significant rework.

Authors Reply: We would like to thank the reviewer for his/her time in reading and critiquing our manuscript, and for providing valuable feedback.

General remarks:

1. Most of the manuscript mentions C2 symmetry. However, in line 64 of the manuscript, also the z-mirror symmetry is mentioned. A definitive statement of the symmetries that must be broken for the described effect to occur and a figure panel illustrating these symmetries would

help my understanding.

Authors Reply: We thank the reviewer for pointing out this problem. We apologize for the unclear interpretation upon the symmetries required for the STOV generation. In fact, as we derived in the SI, the conditions of creating the zero-valued singularity point in transmission are as followed:

$$\begin{cases} (\omega - \omega_0)^2 = \frac{4r^2\gamma_2\gamma_4}{t^2} - (\gamma_2 - \gamma_4)^2 \\ \gamma_1 = \gamma_2 \\ t \neq 0 \end{cases} \quad (\text{S21})$$

In our case, the condition S21.3 ($t \neq 0$) always holds. The condition S21.1 describes the coupling between the background and the radiation from the resonance itself. It's obviously that in the premise of the frequency deviation, the condition S21.1 can also be fulfilled: for a specific resonance at k_x , the mismatch between the background and the radiation can be compensated by the frequency detuning $\omega - \omega_0$. Moreover, considering that we have $r^2 + t^2 = 1$, the degree of freedom (DoF) of the condition S21.1 is exactly 1. Therefore, for a specific structure with given parameters, we can always find a proper deviation ω to compensate the mismatch.

The condition S21.2 describes the requirement of the system symmetry, where γ_1 and γ_2 are radiation decay rates of port 1 (resonances at $-k_x$) and port 2 (resonances at k_x), respectively, shown in Fig. 1d in the main text. Obviously, condition S21.2 asks the system must have the same decay rates from the two radiation channels at $\pm k_x$, which can be fulfilled automatically by preserving the in-plane C_2 symmetry (two-fold in-plane rotational symmetry) or the z-mirror symmetry (vertical mirror symmetry, denoted as σ_z). Because for the z-mirror symmetry, there have $\gamma_1 = \gamma_3, \gamma_2 = \gamma_4$. Otherwise, reciprocity then requires that $\gamma_1 + \gamma_3 = \gamma_2 + \gamma_4$, so that $\gamma_1 = \gamma_2 = \gamma_3 = \gamma_4$. In other words, for system with C_2 in-plane symmetry or z-mirror symmetry, the condition S21.2 holds for any k_x . In this case, the DoFs of singularity is 1 (determined from the condition S21.1), and in 2D parameter space (ω, k_x) , the trajectories of transmission singularity must be a curve close to the energy band: for any k_x , condition S21.2 and S21.3 automatically hold, and a proper frequency deviation ω is determined from the condition S21.1, shown in Fig. 1d in the main text.

However, such a nodal line can't provide the isolated singularity to generate a STOV. To achieve that, we break the in-plane C_2 symmetry and z-mirror symmetry. As a result, the

condition S21.2 doesn't hold constantly. Since both γ_1 and γ_2 are real numbers, the DoF of condition S21.2 turns to be 1, too. Therefore, without the C_2 symmetry and z-mirror symmetry, considering both condition S21.1 and S21.2, the overall DoFs of the transmission singularity should be 2, and thus in 2D parameter space (ω, k_x) we can only find one point satisfying all the conditions: that's the STOV we want. Notice that the condition S21.2 holds at Γ point ($k_x = 0$) due to the protection of reciprocity even without the C_2 in-plane symmetry and z-mirror symmetry. Also, the break of C_2 symmetry and z-mirror symmetry ruin the symmetry-protected BIC at Γ point, open the vertical channel. Therefore, the STOV usually appears at Γ point without C_2 symmetry and z-mirror symmetry, as shown in Fig. 1e in the main text.

As a summary, both C_2 asymmetry and z-mirror asymmetry are necessary to create the STOV: to break the Γ -BIC and to extract the isolated singularity from the nodal line. In the realistic situation, we employ the slanted grating to break the C_2 symmetry and z-mirror symmetry. To eliminate the misunderstanding, we revise the sentence where appears C_2 symmetry. The defining description of C_2 symmetry has been added in line 48-49. Some supplementary descriptions have also been added to line 95-126 of the supplementary material

2. Showing the reference pulse spectrum and spectral phase and correcting for this phase is required. Showing the reference pulse spatial profile would also help evaluate the data.

Authors Reply: We agree with the reviewer's comment that if the reference pulse carries a complex phase, it may interfere with the linear interferometric measurements employed. Nevertheless, our reference pulse was obtained through spectral shaping of the initial femtosecond pulse with a 10 nm bandwidth Gaussian filter. Since there are virtually no transmissive optical components in the reference arm (only the two beam splitters with a low group delay dispersion of $< 20 \text{ fs}^2$ at 800 nm, UFBS5050 from Thorlabs), the reference pulse remains nearly transform-limited and de-chirped ($\sim 80 \text{ fs}$) before the interferometric measurements. To further confirm this, we comprehensively characterized the reference pulse using a commercial frequency-resolved optical gating (FROG) system (FROGscan, Mesaphotonics). As shown in Fig. R1a, the pulse width of the reference pulse (blue solid line) is $\sim 80 \text{ fs}$ and is very close to a transform-limited pulse (red dashed line); its phase (green solid line) is flat, indicating no noticeable chirp. We also measured the spectrum and phase of the

reference pulse (Fig. R1b), which shows that its spectrum (red solid line) is very close to a Gaussian distribution, and the corresponding phase (green solid line) is also flat. Similarly, the measured FROG trace shown in Fig. R1c indicates that the reference pulse does not possess a noticeable chirp. Therefore, the reference pulse we used will not significantly affect the reconstruction of the intensity and phase of the generated STOV. We have added the relevant discussion and figures to the revised manuscript (line 139) and supplementary materials (Supplementary Section IV and Fig. S3).

Fig. R1. FROG measurement results of the reference pulse. a, Temporal intensity (blue solid line) and phase (green solid line) distribution. The red dashed line represents the transform-limited pulse. **b,** Spectral intensity (red solid line) and phase (green solid line) distribution. **c,** Measured FROG trace.

3. The abstract, introduction, and conclusion praise this paper’s developments as a way toward a fully on-chip integrated system for spatiotemporal light manipulation. However, no part of the manuscript deals with integrated photonics. Furthermore, the required light incidence and grating directions are not compatible with integrated photonics manufacturing techniques. I think this statement is wrong or needs further discussion.

Authors Reply: Thanks for the comment. We agree with the reviewer. According to reviewer’s suggestion, the relevant description of on-chip has been removed from the revised manuscript.

4. An in-depth proofing of the article usage in the manuscript, adding explanations for specialized terms, and longer explanations for mathematical deductions would help my understanding.

Authors Reply: Thanks for your constructive suggestions. In order to understand the work

more easily, we have added longer explanations to some special terms and mathematical derivations in the revised manuscript, for example, C_2 symmetry (line 48-49), the Γ point (line 62), the interaction between the pulse and the metasurface (line 68-81), the conditions of singularity in complex transmittance (line 82-97), time diffraction effect (line 166-169), spectral measurement (line 338-343), and so on, see the yellow highlight of revised manuscript and supplementary materials for details.

Specific remarks:

Line 24: ‘frequency-momentum space to the spatiotemporal domain’: I believe the order ‘momentum-frequency space to match spatiotemporal domain’ would be more correct.

Line 31: superfluid is maybe missing a noun.

Authors Reply: We thank the reviewer for pointing out these problems. They have been corrected to line 24 of the revised manuscript.

Line 47: is a slanted metasurface necessary, or could one use other ways to break symmetry (e.g., how more common metasurfaces realize BICs) and still obtain the same effect?

Authors Reply: Thanks for the question. As we replied in the general remarks above, the slanted metasurface is not necessary for the isolated STOV. And other configurations which break both the C_2 in-plane symmetry and z-mirror symmetry can also result the same effect. For example, we can employ the super lattice with two asymmetric grating in one unit cell to break the C_2 symmetry and z-mirror symmetry, as shown in the Fig. R2:

Fig. R2. An example of the configuration which breaks C_2 symmetry and z-mirror symmetry

Comparing to our design where two parameters (w and θ) should be optimized, there are five

parameters (w , r , d , h_1 and h_2) in the super lattice configuration should be optimized. In order to elaborate the physical nature of the STOV clearly and precisely, we choose to employ the slanted grating as the example in the main text.

Line 48 (reoccurs later): I believe ‘fabricated on Silicon’ should read ‘fabricated of/from silicon’.

Authors Reply: Thanks for this constructive suggestion. It has been corrected to line 59 of the revised manuscript.

Fig. 1a: the coordinate system does not match between panel a and the manuscript.

Fig. 1: the polarization could be indicated with an arrow.

Fig. 1b: Gamma should be marked if it is used in the manuscript.

Fig. 1b: it would be interesting to see at least two unit cells to judge periodicity.

Fig. 1d: l is not introduced in the caption.

Fig. 1d/e: the red rectangle is not discussed in the caption.

Fig. 1d/e: The lower panels would be easier to interpret as two-dimensional false-color plots with axes labels and ticks. The third axis and tilted view add no extra information.

Authors Reply: We are very grateful to the reviewer for these valuable suggestions. These problems have been corrected as shown in the figure R3 below. The corresponding figure 1 and caption have also been updated in the revised manuscript.

Fig. R3. Origin of topological singularity induced by symmetry breaking. **a**, Schematic of nonlocal STOV generation, showing an ultrafast light pulse transmitting through a metasurface that is comprised of a one-dimensional slanted grating. The white arrow indicates the polarization direction of the incident pulse. **b** and **c**, The band structure and Q factors of the TE-mode along the k_x axis at $\theta = 0^\circ$ (red) and $\theta = 15^\circ$ (blue), respectively, with the insets showing the mode profiles in the two unit-cells at the Γ point ($k_x = 0$). **d**, For a C_2 symmetric structure, the complex transmission coefficient s presents only one nodal line in the ω - k_x space and does not show a spiral phase in the frequency-momentum space. **e**, By breaking the C_2 symmetry, an isolated zero-transmission singularity associated with a spiral phase appears at the BZ center near the resonant frequency. Middle panel of **d** and **e**: the amplitude distribution of the transmission coefficient. Lower panel of **d** and **e**: the phase distribution corresponding to the red rectangle region.

Fig. 1d/e, 2 c/d, 5 b/c: the employed non-cyclic colormap suggests a phase discontinuity. A cyclic colormap would mitigate this impression.

Authors Reply: Thanks for the comment. The optical vortex, also known as a screw dislocation or phase singularity can show the helical wave-front around the azimuth with an undefined phase and a nullified intensity at the center. The vortex can be quantified by a number, called

the topological charge, according to how many twists the light does in one wavelength. The non-cyclic color map in the vortex phase distribution is very useful for highlighting the topological charge and is also frequently used in the literature related to vortex or spatiotemporal vortices [1-4], so we have retained them in the revised manuscript.

[1] Y. Zhao, J. S. Edgar, G. D. M. Jeffries, D. McGloin, and D. T. Chiu, Spin-to-Orbital Angular Momentum Conversion in a Strongly Focused Optical Beam. *Phys. Rev. Lett.* **99**, 073901 (2007).

[2] S. W. Hancock, S. Zahedpour, A. Goffin, and H. M. Milchberg, Free-space propagation of spatiotemporal optical vortices. *Optica* **6**, 1547-1553 (2019).

[3] G. Gui, N. J. Brooks, H. C. Kapteyn, M. M. Murnane, and C. T. Liao, Second-harmonic generation and the conservation of spatiotemporal orbital angular momentum of light. *Nat. Photon.* **15**, 608–613 (2021).

[4] M. Yessenov, J. Free, Z. Chen, E. G. Johnson, M. P. J. Lavery, M. A. Alonson, and A. F. Abouraddy, Space-time wave packets localized in all dimensions. *Nat. Commun.* **13**, 4573 (2022).

Line 61: the Gamma point should either be marked in the figure, or one could use $k_x=0$ instead.

Authors Reply: Thanks for the comment. We have added the $k_x=0$ in the main text to mark the Gamma point as shown in line 62.

What is the difference between a second-order Gamma point and a first-order Gamma point?

Authors Reply: Thanks for the comment. In photonic crystals, the Γ point refers to the center of the simplified Brillouin zone. Due to the Bloch boundary condition, the Brillouin zone will fold into the simplified Brillouin zone, causing an overlap of the Γ points. To distinguish these folded Γ points, they are labeled as n-order Γ points based on their respective frequencies, with the lowest frequency designated as the first order, as shown in the Fig. R4 below:

Fig. R4. The position of Γ points

Line 64: see general remarks. Is breaking both symmetries necessary or would breaking one symmetry be sufficient?

Authors Reply: Thanks for the comment. As we stated in the response to the general remarks, to create the isolated STOV, both C_2 symmetry and z-mirror symmetry are necessary to be broken.

Line 70: two staggered Fourier transforms would read more clear

Authors Reply: Thanks for this constructive suggestion. It has been corrected in line 78 of the revised manuscript.

Line 71: I cannot understand the sentence ‘Such a relationship...’. Maybe it requires more explanation.

Authors Reply: Thanks for the comment. The relationship mentioned in line 78 refers to the Fourier transform relation $G(\tau, x) = \mathcal{F}\{g(\omega, k_x)\}$. Here the $G(\tau, x)$ denotes the features of the far-field pulse we observe in time and spatial domain; the $g(\omega, k_x)$ is the transmitted light field in the frequency-momentum space, consisting of two ingredients: the characteristics of resonance and the background. This relationship indicates our main claim: a vanishing point in $g(\omega, k_x)$ (zero-valued singularity point upon transmission) in frequency-momentum space can

be conveyed to a vortex in $G(\tau, x)$ in time-spatial domain through the Fourier transformation. To better interpret this claim, we revised the whole paragraph in line 68-81 of the manuscript.

Line 74: can a singularity be not zero-valued? If not, I would drop the zero-valued (occurs again later in the manuscript).

Authors Reply: Thanks for the comment. In general, singularities contain zeros and poles. In this work, we obtain the singularity by constructing isolated zero transmission point, so here we use the term zero-valued singularity.

Line 78: why do you expect the cancelation of the two contributions? The two could have different strengths. Or are equal strengths guaranteed somehow?

Authors Reply: Thanks for the comment. We first apologize for the misguidance to use the word “cancellation”. As we replied to line 71, indeed we ask for a vanishing point in $g(\omega, k_x)$ to create a vortex in $G(\tau, x)$ that we observed in the time-spatial domain. As we stated in the main text, there are two ingredients contributed to the $g(\omega, k_x)$: radiation and the background. However, they are not simply interference with each other, but couple to each other and ruled by the energy conservation law (see TCMT model in the SI). Therefore, the vanishing point of $g(\omega, k_x)$ is not resulted from the cancellation of the two ingredients simply. According to condition S21.1 that we stated in the response to the general remark, it’s not necessary to ensure that the two ingredients always have the equal strength. In fact, we just optimize the geometric structure of the metasurface to find a working point to fulfill the condition S21.1, and thus realize the vanishing point. Moreover, according to condition S21.1, the deviation of the frequency would compensate the mismatch between the radiation and the background, which further loosens the criteria of the equal strengths and enhances the parameter tolerance. In other words, the vanishing point is robust in the 2D parameter space $(\omega-k_x)$ against the inevitable fabrication errors and we don’t need to apply extra efforts to maintain the equal strengths.

Line 82: do you mean ‘for a system without asymmetry’?

Authors Reply: Thanks for the comment. We apologize for any confusion resulted from our unclear expressions. We have rewritten the whole paragraph to better claim the symmetry

dependence of STOV, as below,

“By employing the two-resonance TCMT model (see Supplementary Section I for details), the conditions of singularity in complex transmittance can be concluded as:

$$\begin{cases} (\omega - \omega_0)^2 = \frac{4r^2\gamma_2\gamma_4}{t^2} - (\gamma_2 - \gamma_4)^2 \\ \gamma_1 = \gamma_2 \\ t \neq 0 \end{cases} \quad (1)$$

which is naturally fulfilled at any k_x with the protection of C_2 symmetry or z-mirror symmetry (see Supplementary Section II for details). As verified, we obtain a nodal line in transmittance spectrum in the ω - k_x space (Fig.1d, middle panel). However, since such singularities are not isolated, they cannot exhibit as a spiral phase in the ω - k_x space (Fig.1d, lower panel) and thus cannot be utilized to generate STOVs.

In order to find an isolated singularity of s to generate a STOV, we turn to the metasurface that breaks the C_2 symmetry and z-mirror symmetry, for instance, using the slanted grating on a low-index substrate as shown in Fig.1e. In order to find an isolated singularity in transmittance to generate a STOV, we turn to a metasurface with broken C_2 symmetry and z-mirror symmetry, for instance, using the slanted grating on a low-index substrate as shown in Fig.1e. At nonzero k_x point, the singularity conditions in Eq. 1 are ruined (typically $\gamma_1 \neq \gamma_2$) so that the nodal line would disappear. However, at the time-reversal invariant momentum $k_x = 0$, condition $\gamma_1 = \gamma_2$ still preserves to result a topologically robust isolated singularity in the ω - k_x space. Confirmed by numerical simulations (Fig.1e, mid panel), a vanishing point of transmissistance emerges at the BZ center with a slight frequency detuning from the eigenfrequency of mode TE-A.”

This modification is shown in line 82-97 of the revised manuscript.

Line 83: r, t should be defined, even if they are used commonly.

Authors Reply: We thank the reviewer for pointing out this problem. The r and t are the Fabry-Perot background reflection/transmission coefficients. The relevant descriptions have been added in line 75 of the revised manuscript.

Fig. 2c: the lower panels would be easier to interpret as two-dimensional false-color plots with

axes labels and ticks. The third axis and tilted view add no extra information.

Fig. 2c: caption says 30° , figure says 20° .

Authors Reply: We thank the reviewer for pointing out these problems. According to reviewer's suggestion, they have been corrected in the figure 2 of revised manuscript, as shown below.

Fig. R5. Design of the metasurface with taking non-radiative loss into account. **a**, By adding an optical extinction coefficient to the material ($k_{Si} = 0.0065$) to represent non-radiative losses, the topological singularity of the transmission coefficient does not vanish but deviate to an off- Γ position in ω - k_x space. **b**, The evolution of the topological singularity in ω - k_x space when the slant angle of grating θ varies from 30° to 2° . **c**, The corresponding complex transmission coefficient distribution in the frequency-momentum space for slant angles $\theta = 10^\circ$, 15° , and 20° . **d**, The amplitude and phase distribution of the transmitted STOV in the spatiotemporal domain after a Gaussian pulse propagates through the nonlocal metasurface with the slant grating angle of $\theta = 15^\circ$.

Fig. 2d: judging from the discussion of the experiment, the tau-x distribution occurs after a 4-f imaging system and in a precise position along the propagation direction. Here, it seems the

distribution occurs immediately after the metasurface. If further propagation was included, this should be mentioned.

Authors Reply: Many thanks for the comment. In this work, the intrinsic topological singularity is induced by C_2 symmetry and z-mirror symmetry breaking of the metasurface and exists in the momentum-frequency space. Therefore, the STOV generation requires neither position alignment nor special propagation, and it appears immediately after the metasurface.

In the experiment, the STOV is also generated immediately after the light pulse passes through the metasurface, but it will rapidly evolve and deteriorate within a very small transmission distance. In order to record and detect the intensity and phase distribution of the STOV in the far field by using time-resolved interference technology, a 4f imaging system along the x direction is introduced into the optical path. Therefore, the STOV generated at the initial moment after the metasurface can be recorded by the CCD located at the focal plane of the cylindrical lens.

In the simulation, it is also verified that the STOV is generated immediately after the light pulse passed through the metasurface. The monitor is positioned 2 μm away from the metasurface. To clarify this, the position of the detector during the simulation has been described in line 115-116 of the revised manuscript.

Line 118: It should either be mentioned that the pulse is compressed, or the pulse's temporal/spectral input phase should be given.

Authors Reply: Many thanks for the reviewer's suggestion. The description of compression pulses has been added to line 114 of the revised manuscript.

Line 118: I expect that the spatial profile of a 3 μm waist at 800 nm wavelength experiences quite the evolution upon propagation. Therefore, the position along the propagation direction at which the beam has a 3 μm waist and the position at which Fig. 2c is recorded should be mentioned.

Authors Reply: Many thanks for the comment. In this simulation, the light source is set to compressed Gaussian pulse light with beam waist of 3 μm and center wavelength of 800 nm,

and the corresponding Rayleigh length can be calculated as $z_R = \frac{\pi w_0^2}{\lambda_0} = 35.3 \mu\text{m}$.

The $3 \mu\text{m}$ waist is set at the initial position of the light source, and the metasurface is placed $0.5 \mu\text{m}$ in front of the light source along the propagation direction. The monitor is positioned $2.5 \mu\text{m}$ away from the light source, which is much less than the Rayleigh length. According to reviewer's suggestion, the position have been described in line 116 of the revised manuscript.

Fig. 3c: line-outs along the dashed lines would enable judging data quality.

Authors Reply: Many thanks for this constructive suggestion. Cross sections of the transmission spectra along horizontal (top) and vertical (bottom) white dashed lines have been added in figure R6(d) below. The figure 3 and caption in the revised manuscript have also been updated accordingly. We also added the corresponding description “the transmission spectra possess a clear minimum centered at $k_x a / 2\pi \approx -0.0105$ and $\omega a / 2\pi c = 0.5$ in the 2D ω - k_x space (Fig. 3d)” in line 134-135.

Fig. R6. Fabrication and characterization of nonlocal metasurfaces. **a**, An optical microscope photograph of the fabricated metasurface. **b**, A tilted-view and a side-view scanning electron microscope images of the sample. **c**, The angle-dependent transmission spectra of the metasurface measured under TE polarized incidence. **d**, Cross sections of the transmission spectra along horizontal (top) and vertical (bottom) white dashed lines in **c**.

Fig. 3c: no details on this measurement are provided.

Authors Reply: Many thanks for the comment. The angle-dependent transmission spectra of the metasurfaces are measured with an angle-resolved spectroscopy microscopic system (IdeaOptics Instruments). The incident light polarization is set to along the y direction (TE). The transmitted light is collected by a 100× near-infrared microscope objective lens with a numerical aperture of 0.95. By applying an aperture in the optical path, the illumination area is limited to be 15 μm × 30 μm approximately. In experiments, the spectra resolution is 0.75 nm,

and the angular resolution is 0.2 degree. These relevant descriptions about spectral measurement have been added in Methods of the revised manuscript, as shown in line 338-343.

Fig. 4: the polarization could be added as an arrow.

Authors Reply: Many thanks for this constructive suggestion. A white arrow indicating the polarization direction of the incident pulse has been added to the figure 4 in revised manuscript.

Fig. 4: the drawn interferometer does not allow zero delay. If mirrors were omitted for the purpose of a cleaner drawing this should be mentioned in the caption.

Authors Reply: We are grateful for the reviewer's constructive reminder. Indeed, we omitted some mirrors in the drawing of the interferometer for clarity, aiming to simplify the illustration to facilitate the readers' understanding of the principle of the interferometric measurement. We have now added a related description "To make the drawing clearer, some mirrors for zero delay are omitted" to the caption of Fig. 4 in the revised manuscript to clarify this omission and prevent any potential confusion.

Fig. 4: the drawn interferometer arms have extremely unbalanced dispersion.

Authors Reply: We appreciate the constructive comment from the reviewer. Indeed, in the signal arm of our interferometer, we introduced a few additional optical components (i.e., a spherical lens, a cylindrical lens, and an objective) to spatially shape the signal pulse before and after modulation, which introduces additional dispersion compared to the reference arm, implying that the signal pulses are indeed chirped. Nevertheless, the chirp of the signal pulses does not influence the accurate reconstruction of the intensity and phase of the generated STOV using interferometric measurements, as the correct reconstruction merely requires the exclusion of complex phase information carried by the reference pulse (which we have demonstrated to be nearly transform-limited in our previous response). We also would like to emphasize that the chirped pulses do not affect the generation of an STOV. In fact, many previous experimental works have used chirped pulses to generate STOVs (e.g., refs. [26] and [27] in the main text), and the successful generation of STOV in our experimental results further substantiates this.

Fig. 4: the system is only 4f-imaging along the x-axis. This should be mentioned in the manuscript.

Authors Reply: We thank the reviewer for pointing out this problem. We have added this description about “4f-imaging system along the x-axis” in line 143-144 of the revised manuscript.

Line 140: charactering should probably read characterizing.

Authors Reply: Many thanks for pointing out this spelling mistake. It has been corrected in line 137 of the revised manuscript.

Line 142: same as before: it should either be mentioned that the pulse is compressed, or the pulse’s temporal/spectral input phase should be given. Just giving a pulse duration and no information about the phase does not characterize the laser pulse. This should be accompanied by the measured spectral or temporal phase, using, e.g., FROG, SPIDER, or a related technique.

Authors Reply: Thanks for the comment. Please refer to our previous response regarding the reference pulse spectrum and spectral phase, where we have detailed the spectral shaping and characterization of the reference pulse. We have also added the relevant discussion and figures to the revised manuscript (line 139) and supplementary materials (Supplementary Section IV and Fig. S3).

Line 146: the system is only 4f-imaging along the x-axis.

Authors Reply: We thank the reviewer for pointing out this problem. It has been corrected in the revised manuscript.

Line 168: Is a 90° turn between the spatiotemporal profiles $z=-45\text{mm}$ and $z=45\text{mm}$ indicative of a symmetric or an antisymmetric time evolution?

Authors Reply: We thank the reviewer for their attention to detail. The 90° turn observed between the spatiotemporal profiles at $z = -45 \text{ mm}$ and $z = 45 \text{ mm}$ indeed is the symmetric time evolution, as their intensity distributions are mirror images of each other along the τ -axis.

Line 170: I am unfamiliar with the term ‘pure time diffraction’, maybe an explanation would help.

Authors Reply: We apologize for the confusion caused by our use of the term. The correct term we intended to use is "time diffraction." The time diffraction effect describes the spontaneous evolution of STOVs during free-space propagation, resulting from the accumulation of different phase shifts by various temporal frequency components within the STOV. This effect is the underlying mechanism for the observed symmetric time evolution. Actually, the symmetric time evolution of STOVs has been studied in several established literatures (see refs. [17], [22], [27] in the main text). We are thankful to the reviewer for bringing this to our attention. In the updated manuscript, we have corrected "pure time diffraction" to "time diffraction" and have provided an appropriate explanation for this term in line 166-169.

Lines 171-177: For my interpretation of the term non-local, this line of reasoning does not prove non-locality. It proves that the metasurface/grating is the same in multiple spots. If I look at multiple spots in the same grating, I expect the same performance. A strong non-locality could maybe be shown for different-sized metasurfaces or different input beam waists?

Authors Reply: Thanks for the comment. We agree with your opinion that the experiment results we presented in Fig. 5c in the main text prove the multipoint consistency directly, not the non-locality. However, we still think that the experiment results can reflect the non-locality to some extent. In fact, the multipoint consistency of our metasurface exactly comes from the non-locality. For the inhomogeneous local metasurface where the individual cells differ from each other and the tight-binding approximation can be applied, obviously the multipoint consistency doesn't exist, and the coupling effect is also local around the individual cells. Differently, our structure is periodic and transversely homogeneous. As a result, the coupling effect in our structure is long-ranged and the response is momentum-dependent -- the nonlocal response allows intrinsic topological singularity exist in the momentum-frequency space, and finally convey to the STOV in time-spatial space through Fourier transformation. This indicates that the STOV generation is not dependent on specific local excitation upon the metasurface, verified by our experiment results shown in Fig. 5c. Therefore, we still think that the discussions

in line 170-177 can prove the non-locality of our structure.

Fig. 5a: I see forks. Data evaluation would be easier if a zoomed version of a fork and the reference and signal spatial profile without interference were also shown. Furthermore, understanding the displayed data would maybe benefit from giving a simulated (perfect) example of how the data should look.

Why are there fringes at $\tau=0$ in the y-direction? Is one wave impinging at an angle?

Authors Reply: We appreciate the constructive feedback from the reviewer. As shown in figure R7 below, we have included a zoomed version of the fork and the reference and signal spatial profiles without interference to facilitate data evaluation. In addition, as shown in figure R8, we have also provided simulated interference patterns in the x-y plane at different delay times to help readers better understand the relationship between the unique intensity and phase distributions of the STOV and the resulting interference patterns. We have added these figures to the revised supplementary materials.

Fig. R7. Spatial profiles of pulse light. **a**, Signal pulse. **b**, Reference pulse. **c**, Interference pulse. The insert: zoomed version of the orange dashed rectangular region. Scale bar: 1 mm.

Fig. R8. Theoretical fringe patterns in the x-y plane at various temporal locations of the STOV.

In response to the reviewer's question about the presence of fringes at $\tau = 0$ in the y-direction, we clarify that in our interferometric measurements, the reference and signal pulses overlap while maintaining a slight angular tilt in the x-direction. This orientation results in the observation of interference fringes in the y-direction, and the fringe spacing can be optimized by fine-tuning this angle. Notably, at $\tau = 0$, the measured STOV carries exactly opposite phases across the x-direction, leading to the interference fringes appearing to be displaced by half a period in the x-direction. We have added a related description to line 352-354 in the Methods section of our updated manuscript to provide clarity: "Note that the overlapping reference and signal pulses maintain a slight angular tilt in the x-direction, and we fine-tuning this angle to optimize the period of the observed interference fringes in the y-direction."

Methods:

Line 345-350: There is too little detail to understand or evaluate the phase extraction procedure.

Authors Reply: We apologize for not providing sufficient detail on the phase recovery process in our original manuscript. Actually, the method we employed for extracting the phase of the

STOV is similar to that described in refs. [24] and [27] of the main text. More specifically, for each time delay τ , we apply a one-dimensional Fourier transform along the y-axis to the interference patterns. Therefore, the one-dimensional phase profile $\phi(x)$ at each τ could be obtained by extracting the phase at the peak in the Fourier domain. We then stack the phase profiles obtained at various time delays to construct the complete phase $\phi(x, \tau)$. It is important to note that instabilities in the interferometer introduce additional frame-to-frame noise between the phase profiles at different time delays. To mitigate this issue and obtain the correct phase information, we carefully apply a series of phase shifts to the phase profiles at each time delay. The objective of these phase shifts is to maintain a constant phase value at a reference point x_0 across all profiles, i.e., $\phi(x_0, \tau) \equiv \text{constant}$ for all τ . We have included this detailed description to line 357-365 in the Method of updated manuscript. We believe it provides enough detail for the readers to understand and evaluate our phase recovery process.

Supplementary Information:

Line 35: are r , t just numbers? If they are reflection/transmission coefficients they should be called like that.

Authors Reply: We thank the reviewer for pointing out this problem. The r and t are the Fabry-Perot background reflection/transmission coefficients. The relevant descriptions have been added in line 35-36 of the supplementary materials.

Line 69: how does s relate to S_{14} ?

Authors Reply: Thanks for the comment. In this work, S is the scattering matrix of device. S_{14} is an element of the first row and fourth column in the matrix S , and therefore represents the transmission coefficient from port 4 to port 1. s is a general representation of the complex transmission coefficient of the device, so they are numerically equal.

Line 70/84: writing the complex equation like this adds no information.

Authors Reply: Thanks for the comment. On the one hand, this formal derivation helps to understand that the zero-valued singularity exhibits phase vortex of topological charge $l=1$ in the transmission of metasurface. On the other hand, the complex equations in this section are

also necessary to derive equation (S21).

Line 82: a bracket is missing

Authors Reply: Many thanks for pointing out this omission. It has been added to equation (S19) of supplementary materials.

Line 86/88: are $-2\pi/-1$ also an option?

Authors Reply: Thanks for the comment. The $\Psi = \oint d\varphi$ is the accumulated phase along any closed path \mathbb{C} . $l = \Psi/2\pi$ is the winding number of the path \mathbb{C} around the singularity, also known as topological charge number. In order to clearly illustrate such a relationship, the equation (S20) in the supplementary material has been revised to $l = \frac{\Psi}{2\pi} = 1$.

Reviewer #2 (Remarks to the Author):

The authors report the observation of spatiotemporal optical vortices (STOV) enabled by symmetry-breaking nonlocal metasurfaces. Recently there have been strong interests in spatiotemporal optical vortices due to the transverse orbital angular momentum carried by these novel optical fields. The authors demonstrate the feasibility of generating STOV using a compact device called meta surface. The authors use a standard interferometric method to measure the generated STOV. Given the rapidly increasing researches in STOVs, such a device would be of interests to researchers in the field. However, there are a few points need to be addressed to justify its significance to be published in a high profile journal like Nature Communications.

Authors Reply: We would like to thank the referee for reviewing the manuscript and giving positive comments.

1. The concept of generating STOVs using integrated devices is not new, for examples, references 35 & 38. Although these references did not report experimental realization using the proposed structures, it seems the essential concept has been revealed. Thus it is necessary for the authors to clearly identify the advantages/differences of the proposed meta surface.

Authors Reply: Many thanks for this constructive comment. In the references 35, the authors propose the generation of 3D linear light bullets propagating in free space using a single nonlocal optical surface. This approach can generate the space–time coupling and can indeed produce a vortex of pulse light, but we also notice that the direction of orbital angular momentum (OAM) is along the direction of light transmission. In contrast, the spatiotemporal optical vortices (STOVs) generated by our nonlocal surface are featured by transverse OAM that is perpendicular to the pulses' propagation direction. This transverse vortex is induced by C_2 symmetry and z-mirror symmetry breaking of the nonlocal structure.

In the references 38, the authors for the first time employ photonic crystal slabs with transmission nodal lines to create STOVs with arbitrarily oriented (OAM). This approach highlights the novel use of photonic crystal slabs in manipulating optical vortices. However, this paper focuses more on theoretical and design innovations in controlling optical vortices, and ignores the complexity of nanostructures in in fabrication. Inspired by this, our work

proposes and experimentally validates the generation of STOVs using a slanted grating metasurface on a microscale platform. This method stands out for its high efficiency, exceeding 40 % in STOV generation, and its emphasis on simplifying the experimental setup, which is a significant advantage for practical applications. In summary, the references 38 is more theoretical and design-focused, exploring new opportunities with photonic crystal slabs, whereas our work demonstrates practical advantages in experimental simplicity and application efficiency. The relevant descriptions have been added to line 187-190 in conclusion part of the revised manuscript.

[35] C. Guo, M. Xiao, M. Orenstein and S. Fan, Structured 3D linear space-time light bullets by nonlocal nanophotonics. *Light Sci. Appl* 10, 160 (2021).

[38] H. Wang, C. Guo, W. Jin, A. Y. Song, and S. Fan, Engineering arbitrarily oriented spatiotemporal optical vortices using transmission nodal lines. *Optica* 8, 966-971 (2021).

2. Of course, the main technical achievement is the experimental generation of observation of the STOV generated with fabricated device. It appears that the device is just a grating with slant angle. The question is, why the authors call this device as meta surface? In general, meta surface rely on the geometric phase imparted by circularly polarized illumination. It doesn't appear to be the case in this paper.

Authors Reply: Thanks for the comment. You raised a very insightful question. Metasurfaces are a class of artificial surfaces composed of subwavelength-scale structures that can manipulate electromagnetic waves in novel ways. Indeed, the traditional understanding of metasurfaces, as you mentioned, often involves the geometric phase manipulation, particularly with circularly polarized light. However, the field of metasurfaces is evolving rapidly, and the term "metasurface" has been broadened to include a variety of structures that can manipulate electromagnetic waves through various mechanisms, not limited to geometric phase control. The slanted grating metasurface described in the paper is an example of this broader definition. More recently, researchers may use the term "metasurface" for structures that manipulate light at subwavelength scales, innovatively control wave front and exploit other physical phenomena.

The device described in the paper is a slanted grating metasurface, which implies an ultra-

thin and structured surface with subwavelength features that can interact with light pulse. The slanted grating structure in this study is designed to enable nonlocal generation of spatiotemporal optical vortices (STOVs). This is a sophisticated light manipulation technique, which falls within the realm of functionalities typically associated with metasurfaces. Therefore, while the device does not employ geometric phase control via circularly polarized light, it still falls under the broader category of metasurfaces.

3. Although as claimed by the authors that such a device is much more compact compared with the existing pulse shaper based method, the use of such device also have significant drawbacks. The main problem is the lack of flexibility. For example, can the authors generate high order STOVs, or perhaps Bessel STOV with the proposed platform?

Authors Reply: We are very grateful to the reviewers for reviewing our manuscript. We agree with the reviewer that although the device described in this paper is more integrated and compact than the general purpose pulse shaping system based on Fourier transform, the flexibility is limited. Nevertheless, we also envision some possibilities for using the platform to modulate STOVs, especially to generate higher-order STOVs. Firstly, we note that the transverse orbital angular momentum (OAM) direction of the STOVs can be conveniently controlled using a few simple operations on this platform. For instance, rotating the metasurfaces 180° around the z-axis can reverse the transverse OAM direction. In addition, benefiting from the design advantages of ultrathin planar structures, the topological charge of the STOVs can be decreased or increased by cascading two metasurfaces in either the forward or reverse direction, as shown by the simulated transmission coefficient distributions of the reverse and forward cascaded metasurfaces in Figure R9. We believe that our work has the potential to promote the integration of ultrafast optics and integrated optics and create new opportunities for the study and application of spatiotemporal light fields. The relevant descriptions have been added to line 190-196 in conclusion part of the revised manuscript.

Fig. R9. Controlling STOVS with reverse and forward cascaded metasurfaces. **a**, Schematic diagram of the reverse cascaded metasurfaces. **b**, Simulated transmission coefficient distribution of the reverse cascaded metasurfaces with slant gratings $\theta=15^\circ$. **c**, Phase distribution (topological charge $l=0$) corresponding to the red rectangle region of panel **b**. **d**, Schematic diagram of the forward cascaded metasurfaces. **e**, Simulated transmission coefficient distribution of the forward cascaded metasurfaces with slant gratings $\theta=15^\circ$. **f**, Phase distribution (topological charge $l=2$) corresponding to the red rectangle region of panel **e**.

REVIEWER COMMENTS

Reviewer #1 (Remarks to the Author):

Second Review for "Observation of spatiotemporal optical vortices enabled by symmetry-breaking nonlocal metasurfaces" by Huo et al.

I thank the authors for the extensive clarifications in their response. The authors have added the requested data as figures S3 and S4 and the manuscript now supports the findings. The authors have also provided a comparison with simulations that allows a better understanding of the measured data as Figure S5 and details about their experimental and data analysis methodology. I appreciate the added explanations of the required symmetries and can now follow the manuscript much more easily.

The use of slanting in this context is a cool concept and I recommend publishing the manuscript in Nature Communications in its current form.

Specific Remarks

Fig. S5 could be mentioned for comparison in the caption of Fig. 5.

I am not questioning nonlocality, but I still believe the measurement described from line 170 onwards does not prove nonlocality. This disagreement should not stand in the way of publication.

Reviewer #2 (Remarks to the Author):

I would like to thank the authors to take time to address my comments. My concerns have been partially answered. However, there are still questions remained.

1. The authors explain that the slanted angle grating can be regarded as general type meta surface. That almost means that any type of grating (particularly sub wavelength gratings), slab photonic crystal and most of diffractive optical element (DOE) can be called meta surface. I believe this is really not the case and strongly advise the authors to call this device something else.

2. In terms of the flexibility, the authors made an attempt to use cascaded devices to cancel or increase the topological charge. These were shown through numerical simulation, and I suspect experimental realization would be very challenging. For the case to create topological charge of 2, why not just use the same device with twice grating thickness?

Response to the referees' comments:

Reviewer #1 (Remarks to the Author):

Second Review for “Observation of spatiotemporal optical vortices enabled by symmetry-breaking nonlocal metasurfaces” by Huo et al.

I thank the authors for the extensive clarifications in their response. The authors have added the requested data as figures S3 and S4 and the manuscript now supports the findings. The authors have also provided a comparison with simulations that allows a better understanding of the measured data as Figure S5 and details about their experimental and data analysis methodology. I appreciate the added explanations of the required symmetries and can now follow the manuscript much more easily.

The use of slanting in this context is a cool concept and I recommend publishing the manuscript in Nature Communications in its current form.

Authors Reply: We would like to thank the reviewer again for taking the time to comment on our manuscript.

Specific Remarks

Fig. S5 could be mentioned for comparison in the caption of Fig. 5.

Authors Reply: We are very grateful to the reviewer for the valuable suggestion. We have added the relevant descriptions in line 159 of the revised manuscript as “The experimentally observed phase behavior agrees very well with theoretical fringe patterns (Fig. S5).”

I am not questioning nonlocality, but I still believe the measurement described from line 170 onwards does not prove nonlocality. This disagreement should not stand in the way of publication.

Authors Reply: Thanks for the comment. We agree with your opinion that the experiment results we presented in Fig. 5c in the main text prove the multipoint consistency directly. For the sake of disambiguation, the description of line 171 has been changed as “Because the singularity is in frequency-momentum space and the structure is almost homogeneous, we expect that the incident pulse does not need to be accurately aligned to.....”.

Reviewer #2 (Remarks to the Author):

I would like to thank the authors to take time to address my comments. My concerns have been partially answered. However, there are still questions remained.

Authors Reply: We would like to thank the reviewer again for the valuable feedback, and apologize for not fully addressing reviewer's concerns.

1. The authors explain that the slanted angle grating can be regarded as general type meta surface. That almost means that any type of grating (particularly sub wavelength gratings), slab photonic crystal and most of diffractive optical element (DOE) can be called meta surface. I believe this is really not the case and strongly advise the authors to call this device something else.

Authors Reply: We are very grateful to the reviewer for the constructive suggestion. We agree with your opinion and apologize for the inaccurate use of the term metasurface. In the revised manuscript, the name of the device has been changed from "metasurface" to "slanted nanograting".

2. In terms of the flexibility, the authors made an attempt to use cascaded devices to cancel or increase the topological charge. These were shown through numerical simulation, and I suspect experimental realization would be very challenging. For the case to create topological charge of 2, why not just use the same device with twice grating thickness?

Authors Reply: Thank you for your insightful comment. We agree with your opinion that the experimental realization of a cascade configuration may be difficult in realistic situation. However, by employing a bilayer approach, we can realize similar configuration within a single device, akin to the cascade concept. This is feasible in both fabrication processing and experimental realization. Specifically, as illustrated in Figure R1, we first fabricate a single-layer slanted grating, as stated in the SI (Fig. R1a). Then, the single-layer structure is filled within the low-refractive index glue, such as SU8 (Fig. R1b), after which silicon is deposited using techniques like PECVD (Fig. R1c). Subsequently, the deposited silicon layer is etched through the same etching process (Fig. R1d), and the requisite double-layer grating structure is obtained. Employing this methodology, we successfully achieve a configuration similar to the

cascade gratings but within a single device. The simulation results of this double-layered structure is presented in Figure. R2, where a STOV carrying a topological charge of 2 is clearly obtained. Moreover, leveraging the existing processes, which already facilitates the superposition of multiple layers, allows for the construction of higher-order topological charges using this approach. The related discussion has been added in line 180-186 of the revised manuscript. The corresponding figure S7 in the supplementary material has also been updated.

Figure.R1 The schematic diagram of fabrication

Figure.R2 **a**, Schematic diagram of the double-layer slanted grating. **b**, Simulated transmission coefficient distribution of the double-layer slanted grating. **c**, Phase distribution (topological charge $l=2$) corresponding to the red rectangle region of panel **b**.

As for the suggestion the reviewer gave, in fact, high-order STOV corresponds to high-order winding numbers carried by singularities in the k_x - ω parameter space, and has less

dependence on the grating thickness. To prove this, we performed a simulation upon a grating with twice thickness, shown in Figure.R3. Obviously, the topological charge of the singularity is still one, the same with the case shown in main text.

Figure.R3 The result of the same device with twice grating thickness. **a**, Schematic diagram of the structure. **b**, Simulated transmission coefficient. **c**, Phase distribution (topological charge $l = 1$).